# Model-Based 3D Contact Geometry Perception for Visual Tactile Sensor

**DOI:** 10.3390/s22176470

**Published:** 2022-08-28

**Authors:** Jingjing Ji, Yuting Liu, Huan Ma

**Affiliations:** State Key Lab of Digital Manufacturing Equipment and Technology, Huazhong University of Science and Technology, Wuhan 430074, China

**Keywords:** visual-tactile sensor, model-driven, deformation reconstruction, texture detect

## Abstract

Tactile sensing plays an important role for robots’ perception, but the existing tactile technologies have multiple limitations. Visual-tactile sensor (VTS) is a newly developed tactile detector; it perceives the contacting surface shape, or even more refined texture, by way of the contact deformation image captured by a camera. A conventional visual perception is usually formulated as a data processing. It suffers issues of cumbersome training set and complicated calibration procedures. A novel model-based depth perceptual scheme is proposed where a mapping from the image intensity to the contact geometry is mathematically formulated with an associated tailored fast solver. The hardware calibration requires single image only, leading to an outstanding algorithmic robustness. The non-uniformity of the illumination condition is embodied by the stereo model, resulting in a robust depth perception precision. Compression tests on a prototype VTS showed the method’s capability in high-quality geometry reconstruction. Both contacting shape and texture were captured at a root-mean-square error down to a sub-millimeter level. The feasibility of the proposed in a pose estimation application is further experimentally validated. The associated tests yielded estimation errors that were all less than 3° in terms of spatial orientation and all less than 1mm in terms of translation.

## 1. Introduction

Tactile sensing in fingers plays an important role in the identification and manipulation of objects for robots. Among the current technologies in gaining a tactile perception, visual-tactile sensor (VTS), a newly developed camera-based tactile detector as illustrated in Figure 1a, is now gaining increasing research interest [1,2,3] for its remarkable advantages of high precision, low cost, ease of operation, and its inherent information fusion possibility with vision technologies. It has been deployed successfully in multiple robotic manipulation scenarios, e.g., texture recognizing [4], defect monitoring [5], hardness estimation [6], slip detection [7], and object pose estimation [8]. Figure 1b shows a typical VTS consisting of a camera system, illumination components, and an elastomer assembly. When in contact with a target, the deformed elastomer is imaged capturing the illuminated contact depth features, as shown in Figure 1c. A contact depth perception obtaining Figure 1d could be then carried out to figure out the contact load and shape distribution or even some refined surface texture. In contrast to vision-based depth sensing, the VTS senses deformation by means of a camera, but is completely independent of ambient light. This is due to the reflective membrane film on the upper surface of the elastomer of the sensor and the opaque sensor enclosure, which isolates the outside world and makes the sensor cavity form a closed light environment, so that the contact deformation and photometric information form a stable correspondence. The three-dimensional (3D) deformation is essentially reconstructed from two-dimensional (2D) images.

For each hardware VTS, there is a geometry perceptual scheme which is associated with it in determining the hardware illumination and fulfilling the contact depth perception from the elastomer images captured. The basic principle employed in a VTS perceptual scheme falls in photometry-oriented theories. Traditionally, the implementation of a 3D VTS geometry perceptual scheme is data driven. A great amount of image data is required in calibrating the VTS and establishing the perceptual mapping model. The core methodologies could be categorized into photometric stereo mapping and marker-based mapping. A typical case of photometric stereo mapping is demonstrated in [9], in which a look-up table was built in a calibration phase recording the mapping between RGB pixel value and the contacting surface gradients. Some RGB tricolor light sources were configured in advance. A multigrid fast Poisson solver was then employed for reconstructing the depth field to a pixel-wise resolution. The photometric stereo mapping method has been utilized successfully in various of GelSight-type VTSs, e.g., GelSight [10] and GelSlim 3.0 [11]. Yuan et al. [10] estimated the surface normal on each pixel of the shaded image by combining the shading from three or more directions, and then integrated the surface normal for obtaining the 3D shape of the surface. Taylor et al. [11] optimized the shape of the elastomer for improving the uniformity of illumination, so as to construct the mapping accurately. Furthermore, to deal with the non-uniform illumination condition on the sensing surface, Li et al. [12] proposed a deep-learning method for compensating artifacts in mapping process, in which both RGB pixel-wise triples and the pixel positions were fed into ConvNets for estimating the surface. Recently, Wang et al. [13] adopted multi-layer perceptron (MLP) in obtaining a mapping for producing continuous results using a compact GelSight-type VTS named GelSight Wedge. Additionally, Sodhi et al. [14] revealed that the MLP method produces high mean square error in a commercial GelSight-type VTS named Digit [15] owing to its non-ideal illumination.

Benefitting from the development of microstructure forming technology, a method exploiting the algorithms of intuitive feature mapping, namely, marker-based mapping, has been developed. In such a method, arrayed markers are attached onto the elastomer to find the contact depths hinted by the pattern of the marker array. Yu et al. [16] reconstructed the contact depth based on an array of ribbon-shaped markers with the help of Fourier-Transform Profilometry. Lin et al. [17] embedded a dual-layer colorful marker array into a transparent elastomer. A linear relationship was established between the normal displacement and the mean hue degree introduced by the moving markers. Du et al. [18] created a pixel-wise random color pattern as the tracking target, and the contact depth map could thereby be estimated by observing the Gaussian density distribution feature of 2D displacement field. Yang et al. [19] employed a piecewise integral schedule over the gradients of the markers’ displacement so that the contact geometry could be obtained.

Various hardware factors have substantial effects on the outcome of a photometry-based 3D depth perceptual scheme, including the light locations, the uniformity of illumination, and the appropriateness of the light intensity. This sets up a demand for carefully fabricated and tuned hardware of a VTS. Moreover, a data-driven perception requires quite a large bunch of calibrated data for training. Sferrazza et al. [20] used a neural network to estimate the normal force in a novel VTS, in which 13,448 experiments were conducted in building the training set. Rares et al. [21] leveraged three popular deep learning architectures in reconstructing the contact depth. A total of 21,960 training images were used in which various geometries were covered. An off-the-shelf encoder–decoder network frame with a pre-trained image-depth model was tailored into an anthropomorphic hemispherical-shaped sensor [22], by which the restriction on dimensions of a traditional look-up table could be circumvented. One image would be sufficient for a precise depth reconstruction by their network. In total, 29,200 training configurations were consumed. In this way, a problem for such a VTS application is that one has to make sure that the trained network strictly coincides with the hardware situation. Any subtle deviation in the sensor dimensions or materials would mean the calibrated and trained model was wasted. The tedious calibration process itself makes the job even more complicated.

In general, the conventional data-driven depth perception is often implemented as a series of image-processing schemes where the calibration for the lighting environment is usually sophisticated, and a giant bunch of training data is consumed for approximating a photometric model in an inversed style, and thus, it is usually a tedious routine. To address this issue, a near-field photometric model which has been recently established theoretically is considered for bridging the gap between the image and the depth map in a forward direction, eliminating the requirement for the large amount of training data. A novel model-driven depth perceptual scheme for VTS is thereby proposed in this paper in which a mapping from the image intensity to the contact geometry is mathematically formulated with an associated tailored fast solver. The contact geometry could thereby be determined quickly by performing the reconstruction online in a compact matrix form, given that the lighting condition had been kept offline in the database. This novel scheme requires only one image, taken in a non-contact free situation, for calibrating the sensor lighting condition, free of data-expensive calibration or learning procedures. In terms of hardware development, a configuration is deployed in which the LED array is placed so that the light incidents to the elastomer obliquely, leading to an improved almost-no-shade view field. A VTS prototype was designed and fabricated, with a compact structure and a high-quality illumination. A 3D contact geometry perception of the target surface and an estimation of the target’s pose were then experimentally conducted on the prototype sensor.

## 2. Three-Dimensional Geometry Perceptual Scheme

Figure 1a demonstrates a VTS [23] attached to a robot arm’s fingertip in grasping and manipulating an object. The schematics of the VTS are depicted in Figure 1b, where the configuration includes a camera, an LED array with the R, G, and B colors located evenly at 120° apart, a transparent elastomer coated with a reflective membrane, and a transparent acrylic plate for supporting. The light emitted from an LED is homogenized by a diffuser and then reduced in its intensity by a filter, before its oblique incidence into the acrylic plate and the elastomer. The light then returns diffusively at the reflective membrane into the inner cavity of the sensor, forming a uniform light intensity illumination. A contact with an object causes a deformation of the elastomer, which in turn changes the illumination in the cavity. When the sensor is working, colorful images of the internal illumination showing the contact are captured by the camera, see Figure 1c. A prescribed near-field photometric stereo model is then leveraged, by which a geometrical depth perception in the contacting area is conducted upon each clear RGB image frame, see Figure 1d. No intrinsic calibration is required for the camera throughout these procedures. An image coordinate system is established on the image plane of the camera, with its origin O being located at the upper-left corner, the positive X axis pointing vertically down, and the positive Y axis pointing horizontally right, as depicted in Figure 1c. The corresponding sensing area of the VTS is delimitated by the dashed box shown in Figure 1b. The reconstructed depth map Z (X,Y), which is essentially the depth value calculated at each pixel indicated as a pair (X,Y) on the image plane, provides a basis for the perception of the contacting shape and texture. The depth coordinate Z together with the image coordinate system forms a depth coordinate system OXYZ, as in Figure 1d. Once a contact area is identified, a corresponding contact coordinate system O_c_X_c_Y_c_Z_c_ is established as shown in Figure 1e, where the origin O_c_ is set at the centroid of the sensing area, and the positive Z_c_ axis parallels with the positive Z axis. The axes X_c_ and Y_c_ both stay perpendicular to Z_c_ following the right-hand rule. However, the exact directions of X_c_ and Y_c_ are determined at the operation convenience of the application in which the VTS is deployed.

### 2.1. Formulation

Photometric stereo algorithm is a procedure designed to figure out the 3D surface shape, usually referred to as a depth map, from the array of intensity values in a single 2D color image. In realizing a photometric stereo application, lights from multiple directions are necessary to illuminate the contact surface. A camera takes images of the contact surface, providing a hint for the pixel-wise surface normal by the intensity distribution. The final depth map characterizing the contact deformation is the result of a reconstruction step based on a gradient field that follows the surface normal.

Owing to the compact design of a VTS, the light sources are located near the reflective membrane. In such a case, the normal direction $n_{i}$ at a point $P_{i}$ on the membrane (the surface in contact) is connected with the pixel intensity $I_{i}$ imaged at $P_{i}$ by a governing near-field photometric stereo model [24], detailed as Equation (1).
(1)$$\left[\begin{array}{c} I_{i}^{R}\\ I_{i}^{G}\\ I_{i}^{B} \end{array}\right]=\left[\begin{array}{ccc} \rho^{R}\frac{\left(P_{ix}-S_{x}^{R}\right)}{{\left\|P_{i}-S_{}^{R}\right\|}^{3}} & \rho^{R}\frac{\left(P_{iy}-S_{y}^{R}\right)}{{\left\|P_{i}-S_{}^{R}\right\|}^{3}} & \rho^{R}\frac{\left(P_{iz}-S_{z}^{R}\right)}{{\left\|P_{i}-S_{}^{R}\right\|}^{3}}\\ \rho^{G}\frac{\left(P_{ix}-S_{x}^{G}\right)}{{\left\|P_{i}-S_{}^{G}\right\|}^{3}} & \rho^{G}\frac{\left(P_{iy}-S_{y}^{G}\right)}{{\left\|P_{i}-S_{}^{G}\right\|}^{3}} & \rho^{G}\frac{\left(P_{iz}-S_{z}^{G}\right)}{{\left\|P_{i}-S_{}^{G}\right\|}^{3}}\\ \rho^{B}\frac{\left(P_{ix}-S_{x}^{B}\right)}{{\left\|P_{i}-S_{}^{B}\right\|}^{3}} & \rho^{B}\frac{\left(P_{iy}-S_{y}^{B}\right)}{{\left\|P_{i}-S_{}^{B}\right\|}^{3}} & \rho^{B}\frac{\left(P_{iz}-S_{z}^{B}\right)}{{\left\|P_{i}-S_{}^{B}\right\|}^{3}} \end{array}\right]\left[\begin{array}{c} n_{ix}\\ n_{iy}\\ n_{iz} \end{array}\right],$$

Or in a concise matrix format and its inversed version:(2)$$I_{i}=L_{i}\cdot n_{i}=F(S^{j},\rho^{j},P_{i},n_{i}),$$
(3)$$n_{i}=L_{i}^{-1}\cdot I_{i}=F(S^{j},\rho^{j},P_{i})$$
where:
Subscript i∈ℕ: pixel index;Superscript j=R,G,B: color channels;$I_{i}={\left[\begin{array}{ccc} I_{i}^{R} & I_{i}^{G} & I_{i}^{B} \end{array}\right]}^{T}$: imaged intensity triple at the ith pixel;$L_{i}$: linear matrix involves the projected RGB illumination parameters at the ith pixel;$n_{i}={\left[\begin{array}{ccc} n_{ix} & n_{iy} & n_{iz} \end{array}\right]}^{T}$: normal direction of the contact surface at the ith pixel;$\rho^{j}$: reflective albedo of the membrane in the given j color channel;$P_{i}=\left[\begin{array}{ccc} P_{ix} & P_{iy} & P_{iz} \end{array}\right]$: coordinates of the ith pixel to be reconstructed, in terms of OXYZ system;$S^{j}=\left[\begin{array}{ccc} S_{x}^{j} & S_{y}^{j} & S_{z}^{j} \end{array}\right]$: coordinates of an equivalent spot light source of the physical LED array in j color, in terms of OXYZ system.

Equation (1) or (2) embodies a classical photometric model in which each element in the linear matrix $L_{i}$ is described ab initio as a function of the spatial coordinates of point $P_{i}$, the light sources, and the albedo involved in the reflective membrane. However, Equation (3) is the actual model one has to work with in performing a depth perceptual task to find the surface normal given the intensity distribution in an image captured. It is worth noting that the projections of the three mono-color lights are ensured non-coplanar by the sensor design, so that the matrix $L_{i}$ is always full-ranked and thus invertible.

The equivalent lighting location $S^{j}$ involved in both Equation (2) and 1b is defined as the coordinate of a spotlight where the certain luminous intensity is satisfied at a given position. It is worth noting that it is practically infeasible to deduce the location $S^{j}$ of an equivalent mono-color light source by summing up directly the physical coordinates of an array of LEDs of that color. Thus $S^{j}$ is normally different from the physical coordinate of any actual light source. However, $S^{j}$ keeps constant once the VTS is fabricated. It may be calibrated in an optimization approach with the help of single image. The calibration for $S^{j}$ is to be described later.

Then, a reconstruction-from-gradient method is introduced for establishing a mapping from the surface normal $n_{i}$ to the depth $z_{i}$ at that ith pixel. The depth reconstruction could be mathematically reformulated as solving a Poisson equation,
(4)$$\Delta z_{i}=\frac{\partial p_{i}}{\partial x}+\frac{\partial q_{i}}{\partial y},$$
(5)$$\mathrm{where}\;p_{i}=-\frac{n_{ix}}{n_{iz}};q_{i}=-\frac{n_{iy}}{n_{iz}},$$
where Δ is the Laplace operator, $z_{i}$ is the depth at the ith pixel, and $p_{i}$, $q_{i}$ are the gradient components at that pixel in X and Y directions, respectively.

A Poisson equation such as Equation (4) could be attacked by a fast Poisson solver in which Discrete Sine Transform (DST) and Inverse Discrete Sine Transform (IDST) are leveraged. Using of a central differencing scheme in approximating the gradient while running the solver may provide effective control over the reconstruction error.

Thus far, the novel model-driven 3D perceptual scheme from a 2D image intensity is formulated as the sequential combination of Equation (3), the inverse photometric model from intensity to surface normal, and Equation (4), the depth reconstruction from the surface normal. To implement the full perception, one needs to first attack Equation (3) and then Equation (4).

For Equation (3), however, it is impractical to solve directly for the normal $n_{i}$ of the deformed surface from the image intensities. The problems here include the yet unknown illumination parameters $\left\{S^{j},\rho^{j}\right\}$ and the fact that the necessary geometry information $P_{i}$ is unknown and coupled with $n_{i}$. To attack this inverse problem, two tactics are adopted. An offline calibration is tailored for estimating the illumination parameters $\left\{S^{j},\rho^{j}\right\}$, and an approximation scheme is proposed in which $P_{i}$ and $n_{i}$ are decoupled. By doing this, the inverse model Equation (3) could be solved analytically.

Once Equation (3) is solved, achieving the normal $n_{i}$, then the gradient field, the fast Poisson solver could be invoked to attack Equation (4) to find a depth estimation.

The full procedures of the model-driven perceptual scheme proposed are demonstrated in Figure 2, where the offline calibration with a database and the online Poisson-solver reconstruction are stacked. In the offline calibration phase, one image captured under a non-contact situation by the camera is used. The near-field photometric stereo model is employed in analyzing the image to determine offline the illumination condition of the VTS in an iterative optimization way. The a priori parameters of the illumination, including $S^{j},\rho^{j}$, a compensating term $I_{\Delta }$ to be introduced later, and an inversed baseline matrix $L_{i,0}^{-1}$, are thereby obtained and stored in a database for later use. In the online reconstruction phase, a compensation to the raw contact surface images is first conducted using the predetermined correcting term $I_{\Delta }$ in order to obtain a correction to some hardware-caused constant outlier pixels in the intensity distribution. The normal direction at every point of the deformed surface is then calculated from the compensated images with the help of the calibrated a priori illumination knowledge. A gradient field could be readily deduced from the normal directions. Additionally, the depth estimation follows by way of a reconstruction using the fast Poisson solver from the gradient field.

#### 2.1.1. Offline Calibration of Illumination Conditions

The reflective membrane is assumed a Lambertian surface. An image of the non-contacted reflective membrane is taken. With a guess of $\left\{S^{j},\rho^{j}\right\}$, the forward model of the near-field photometric stereo algorithm, i.e., Equation (2), is used for determining a corresponding image intensity guess. The normal displacement $P_{iz}$ is zero here as no contact deformation exists. To push the image intensity guess as close to the actual image as possible, a cost function is defined as the sum throughout the image of the squared error of the image intensity guess $I_{est}$ against the actually captured $I_{init}$. The calibration for the illumination is essential to work out an optimized illumination solution $\left\{S^{j},\rho^{j}\right\}$ by which the cost function is less than the given threshold, shown in Equation (6). In practice, this optimization could be conducted with the help of, e.g., *fminunc* function in MATLAB.

Substituting the solution $\left\{S^{j},\rho^{j}\right\}$ into the near-field photometric stereo model yields an intensity approximation I* to the actual image $I_{init}$. The difference between **I*** and $I_{init}$ is attributed primarily to two fluctuations. One is the random white-noise, which is time-dependent and is normally averaged and close to zero over the sensing area. The other is some intensity outlier pixels which are caused by some occasional hardware deviation or abnormality, e.g., localized spot defect on the membrane, or bad pixel in the camera sensor. The outlier difference is usually constant and not distributed near zero. To get rid of this constant and sometimes obvious difference, a compensating residual distribution $I_{\Delta }$ is introduced as in Equation (7) for restraining the aforementioned constant hardware outlier perturbation in the actual system in order to facilitate the analysis using the theoretical model. This compensation is indeed a correction, or numerical filtering, principled by the model onto the actual image in order to reduce the potential fitting error in deploying the light source model.
(6)$$\left\{S^{j},\rho^{j}\right\}=\arg\min_{S^{j},\rho^{j}}\sum_{P_{i}\in \mathrm{Block}}{\left\|F(S^{j},\rho^{j},P_{i},n_{\mathrm{init},i})-I_{\mathrm{init},i}\right\|}^{2}$$
(7)$$I_{\Delta ,i}=I_{\mathrm{est},i}-I_{\mathrm{init},i}=F(S^{j},\rho^{j},P_{i},n_{\mathrm{init},i})-I_{\mathrm{init},i}$$

With a condition that $P_{iz}$ is zero at all points, substituting the optimized illumination solution $\left\{S^{j},\rho^{j}\right\}$ into Equation (3) gives the illumination baseline coefficient matrix $L_{i,0}$ in which the mapping from the normal to the intensity triples in the non-contact condition is embodied. The subscript 0 indicates the non-contact initial condition which depends only on the hardware implementation of the VTS. The inverse $L_{i,0}^{-1}$ of this baseline is then calculated and stored in a database for later use in the online reconstruction phase. In addition, the reflective albedos at the membrane, and the equivalent locations of the light sources, could be accurately calibrated and stored in the database in advance.

#### 2.1.2. Poisson-Solver Online Reconstruction

To illustrate this phase clearly, a reconstruction instance shown as the five sequential images in the lower row of Figure 2 is taken as a demonstration. From left to right, they are the raw contact image $I_{cont}$, the noise-compensated image I, the contact area identified, the gradient solved in the contact area, and the reconstructed depth map.

Once a raw contact image $I_{cont}$ is obtained from the camera, the outlier compensation is first ignited by applying $I=I_{cont}+I_{\Delta }$. The hardware-defect-corrected I fits closer to an intensity distribution approximated by the calibrated near-field illumination. Then, the contact area is identified and extracted from the compensated I, by way of a binarization, a hole filling, some morphological manipulation, then a subordinate connected-domain removal. The orthogonal gradient fields p and q within the extracted contact area are then obtained using Equation (5), following the normal distribution $n_{i}$ determined from Equation (8). The gradient field outside the identified contact area of an image is set to all zero. With the gradient distribution in hand, the fast Poisson solver is invoked to attack Equation (4) for finding the depth map.
(8)$$n_{i}=L_{i}^{-1}\left(I_{cont,i}+I_{\Delta ,i}\right),$$
(9)$$\mathrm{Fast}\;\mathrm{method}:\;L_{i}^{-1}=L_{i,0}^{-1},\;\mathrm{assuming}\;\mathrm{that}\;P_{iz}=0$$
(10)$$\begin{array}{ll} \mathrm{Precise}\;\mathrm{method}: & \mathrm{Find}\;\mathrm{an}\;\mathrm{initial}\;\mathrm{depth}\;\mathrm{map}\;\mathrm{from}\;L_{i,0}^{-1}\;\mathrm{to}\;\mathrm{be}\;a\;\mathrm{generally}\;\mathrm{non}\text{-}\mathrm{zero}\\ & P_{iz},\;\mathrm{calculate}\;\mathrm{the}\;\mathrm{nudged}\;L_{i}^{-1}\;\mathrm{using}\;P_{iz}\;\mathrm{into}\;\mathrm{Equation}\;(2). \end{array}$$

In determining the normal $n_{i}$ using Equation (8), the inversed coefficient matrix $L_{i}^{-1}$ shall be determined in advance. In the case of a small compression, the inverse $L_{i,0}^{-1}$ of the calibrated baseline matrix from the database is used. Additionally, a 3D depth map is obtained directly in one shot. In this case, the distance from the contact surface to a light source is much greater than the deformation depth. Given that the elastomer surface is nearly a plane, the compression-caused perturbation on the illumination distribution in adjacent to the contact surface is quite small and thus negligible. Thus, the inversed baseline $L_{i,0}^{-1}$ is still a good approximation to the actual coefficient matrix $L_{i}^{-1}$ in Equation (8). In other words, the matrix $L_{i}^{-1}$ could be treated as constant before and after the compression. The matrix $L_{i}^{-1}$ in Equation (8) for this case is thus just a copy of $L_{i,0}^{-1}$ assuming that $P_{iz}$ is zero, as shown in Equation (9). This trick applies as well in the case of a lower precision requirement on the final result in order to leverage its advantage in the operation speed. For the analysis targeting to the error introduced by this zero-$P_{iz}$ assumption, see the following section.

On the contrary, in the case of a relatively heavier compression, the inversed baseline $L_{i,0}^{-1}$ does not apply directly into Equation (8) as the actual compression depth distribution is no closer to the zero-$P_{iz}$ condition implied in calibrating $L_{i,0}^{-1}$. A nudge step updating the calibrated $L_{i,0}^{-1}$ to the applied $L_{i}^{-1}$ is tailored. The inversed baseline is first used to find an initial normal guess. Then, the normal guess is transformed into a gradient field by Equation (5) and fed into Equation (4) in order to obtain a nudged depth guess. By applying the nudged depth guess as $P_{iz}$ into Equation (1), a nudged thus more realistic matrix $L_{i}$ and its inverse could be determined. The normal guess is then re-calculated using the nudged inverse matrix $L_{i}^{-1}$, thus achieving a better estimation of the gradient field and the ultimate depth map.

### 2.2. Error Analysis by Simulation

To evaluate the operational performance of the model-driven depth perceptual scheme developed for a VTS, the error of the calculated depth map from the Z-coordinate ground truth within the contact area is analyzed quantitatively. The primary potential error source of the method presented is two-fold. One is the assumption that the depth $P_{iz}$ is zero in finding an approximation of $L_{i}^{-1}$ for the fast method of Equation (9) The other is the uncertainty introduced in the calibration for the illumination parameters.

Indeed, an error analysis is quite a challenge in a practical test, as the exact Z-coordinate of the contact area is difficult to identify. This difficulty is usually attributed to the geometric uncertainty of the target object and the compression manipulation in controlling the test. The former makes it infeasible to obtain an accurate ground-truth description of the target texture, while the latter introduces some uncertainty to the benchmark for the applied compression depth field. In view of this, the formulation error analysis was conducted in our research by a numerical simulation, in which the reconstruction was carried out at a pixel-wise precision.

#### 2.2.1. Error by the Zero-*P*_*iz*_ Assumption

With a prescribed set of illumination parameters and contact ground truth, a simulation for contact images was conducted applying the photometric model as Equation (3). The perceptual method proposed was then employed to find the depth map. The error was analyzed quantitatively against the contact ground truth in order to verify the appropriateness of this zero-$P_{iz}$ assumption. As the illumination situation was fully given a priori without any uncertainty, the assumption was left the only source of error.

A compressing-sphere case was studied in a MATLAB-based simulation, see Figure 3a. The radius of the compressed sphere was equivalent to 160 pixels. Two compression depths were 16 pixels (depth d1, a shallow-press), and 32 pixels (depth d2, a deep-press). As a simulation in MATLAB, the light source locations $S^{j}$, the reflective membrane albedos $\rho^{j}$, and the surface point coordinates $P_{i}$ known by the sphere geometry, were put into the model Equation (1), as shown in Figure 3a. By this, one obtains the intensity triple $I_{i}$ numerically at every point on the surface. The simulated $I_{i}$ is depicted graphically in Figure 3b, where each subplot is sized 640 × 480. The subplots in Figure 3b, denoting d1 and d2 compressions, were then used as the contact images and fed into the depth reconstruction using Equations (9) and (10) (hardware compensating term $I_{\Delta }$ is zero in this simulation), respectively. The errors in the resulting depth maps of both d1 and d2, that is, the absolute difference between the prescribed depth ground truth and the reconstructed one, are demonstrated in Figure 3c,d, respectively.

For the d1 shallow-press, the maximum reconstruction errors of Equations (9) and (10) were 0.31 pixel and 0.09 pixel, respectively. For the d2 deep-press, the maximum errors of Equations (9) and (10) were 1.21 pixel and 0.16 pixel, respectively. In general, the reconstruction reached fairly precise results in both shallow and deep press cases. This result reveals that the assumption of zero-*P_iz_* in determining the inversed matrix $L_{i,0}^{-1}$ contributes little to the error in the ultimate reconstruction outcome. A deeper compression may cause an increased error. However, the precise method of Equation (10) may help in this case.

#### 2.2.2. Overall Error Performance of the End-To-End Perceptual Method

A contact simulation was conducted using a photometric simulator TACTO in order to validate the end-to-end procedures of the model-driven depth perception schemed in Figure 2. TACTO [25] is an open-source high-fidelity photometric simulator created by Facebook targeting to facilitate the development of VTS. Integrating the *Pyrender* library in Python with the configuration data of a real sensor endows TACTO with rich capacities in simulating the physical contact of an object to the sensor with a high fidelity. In implementing the virtual test, a set of full geometry and illumination data of a VTS was poured into TACTO, making a virtual duplicate of the sensor. In addition, given some parameters describing a contact, a photometry-based simulation was then invoked in TACTO, and a bunch of contact images were rendered. These virtual images stand indeed as some high-fidelity surrogates of the contact image that we could capture in a real VTS. The perceptual scheme proposed was then employed onto the virtual images created by TACTO, trying to recover the contact condition. The perceptual error was then identified as the difference between the reconstructed depth map and the prescribed contact ground truth. The validity of the model-based perceptual scheme proposed could thus be proved synthetically by this simulation.

A non-contact image was firstly generated in TACTO for the estimation of the illumination parameters, as offline calibration in Figure 2. Then, a case in which a radius of 160 pixels (equivalent to 10 mm) sphere was pressed virtually 32 pixels (or 2 mm) into a VTS was studied, obtaining simulated contact images with size 640 × 480, see Figure 4a. In this case, we performed, as suggested, the precise method Equation (10) before the Poisson solving as the final depth reconstruction. The reconstructed result is presented in Figure 4b. As the location of the sphere center and the radius are both a priori known, the ground truth of the contact surface, i.e., the sphere, could be determined theoretically. The absolute error of the reconstruction from the ground truth is shown in Figure 4c. The maximum error presented in Figure 4c was around 3 pixels, or approximately 0.2 mm.

Now proceed one more step forward, to estimate the sphere center and the radius from the reconstructed depth distribution with the help of the least square method. The radius benchmark of the sphere was 10 mm. Additionally, in the contact coordinate system O_c_X_c_Y_c_Z_c_, the ground truth of the sphere center was (0, 0, −8) mm. Their estimated counterpart based on the reconstructed depth were the center at (0.0310, 0.0320, −7.6498) mm and radius of 9.7993 mm, with the relative estimation errors being (+0.3%, +0.3%, 4.4%) and −2.0% for the center and the radius, respectively. This virtual contact experiment on TACTO clearly validates the feasibility of the model-driven end-to-end perceptual scheme proposed in determining precisely the depth distribution in contact.

## 3. Experimental Results and Discussion

A prototype of VTS with its performance briefed aside is demonstrated in Figure 5c. A monocular camera, a transparent elastomer coated with a reflective membrane, a supporting acrylic plate, and RGB light sources with their corresponding diffusers and filters are presented. The LED mono-color light sources of three channels are located 120° apart from each other. The base of each LED is tilted 60° from the horizontal plane. The light from a mono-color LED is first homogenized by a diffuser then reduced in its intensity by a filter, before its incidence into the acrylic plate and the elastomer in an oblique direction. The refraction in the elastomer reduces the angle between the light ray vector and the normal of the reflective membrane to 38.4°. That way, the shading effect in lighting the contact surface is highly attenuated, leading to a better lighting condition, while obtaining a more compact sensor structure. The effective sensing area is dash-boxed in Figure 5b. The material, and the fabrication method or the source of each component forming the prototype, is listed as follows in Table 1. The focus of the camera shall be set up in advance before the prototype assembly. To do this, the transparent elastomer is first placed on the supporting acrylic plate. Additionally, a paper on which some graphs are printed is gently placed on the sensing side of the elastomer. Then, the focus length of the camera is adjusted carefully until the graphs are imaged clearly. The focal length is then locked before the sensor assembly.

A test platform, as shown in Figure 6a, was built for testing the geometry-perception and pose-estimation capability of the perceptual scheme proposed on the VTS. The platform was composed of a tensile and compression testing machine (ZQ-990B), a DC power supply (RIGOL^®^ DPI 308A, China), a connecting flange, a laptop controlling the collection/processing/presentation of all data, and the VTS under test. The tensile and compression testing machine plays as a compressing tester in this platform, providing a capacity of press step distance 0.01 mm in its axial direction with a controlling precision within ±1%. The DC power provides a stabilized voltage for the VTS. A contact target was pin-attached to the flange while conducting an experiment. Additionally, the VTS was thread-fixed on the base, see Figure 6c. Activating an axial movement of the compressing tester caused a contact of the target onto the VTS. The camera captured the contact images at a rate of 30 fps and sent them to the laptop via USB. The model-driven perceptual scheme was invoked online to figure out the depth map at a reconstruction rate of approximately 7.7 Hz. The result was then presented in a contour style.

### 3.1. Contact Shape and Texture Perception Experiments

A set of experiments was conducted on the VTS prototype to validate its capacity in shape and texture perception, by contacting some target parts manually onto the sensor. The targets involved were categorized into a “geometry” group and a “texture” group shown in Figure 7a. The former included ➀ the end of a hand tool and ➁ a cross-head screw. Additionally, the latter included ➂ a textured sphere and ➃ the knurled slider of a utility knife. The contact images of these targets obtained in the VTS are tabulated in Figure 7b. The model-driven perceptual scheme proposed in our research was employed in determining the depth map. A traditional look-up table method [10] was used as well for a comparison purpose. The resulting depth maps from the proposed and the traditional method are demonstrated in Figure 7c,d, respectively. These results are further colorized in a contour style in Figure 7e,f. The colorization of all results is normalized to a unified range to facilitate a qualitative analysis, by which a one-to-one mapping from a color to a depth value is ensured. Due to the difficulty in obtaining the true value of depth in this manual test circumstances, no absolute error analysis was conducted. Instead, we tried to find the difference between the results from the two methods by subtracting the depth result of the look-up table method from that of the proposed model-driven one. Then, root-mean-square errors (RMSE) were calculated using Equation (11),
(11)$$RMSE=\sqrt{\frac{1}{m}\sum_{i=1}^{m}{\left(D_{i}-H_{i}\right)}^{2}},$$
where $D_{i}$ and $H_{i}$ are the 3D depths reconstructed by the model-driven method and by the look-up table method, respectively. The calculated errors include one over the image and the other within the identified contact area. The unit of both depth and error is pixel, where 1 pixel equivalents to 39.5 μm. A quantitative comparison of these difference is listed in Table 2.

The following observations can be made from the experimental results in Figure 7 and Table 2.

Figure 7b gives an impression of a well-established lighting environment inside the VTS prototype, and of the high-contrast contact images captured depicting clearly the shape and texture. This implies the appropriateness of the components of VTS in fulfilling the hardware requisition for the 3D geometry detection. Shape and texture details, including a hole of 1.2 mm diameter on ➀, Philips-head screw with a slot of 1.4 mm width on ➁, protruding ribs of 0.5 mm thickness on ➂, and inter-slot gap of 0.3 mm between particles on ➃, are captured in detail.By a comparison between Figure 7c,d, and that between 7e,f, one could tell that the perceptual result by the proposed model-driven method is quite close to the that obtained by the traditional benchmark method, indicating the ability in an accurate 3D depth reconstruction of our method. The quantitative comparison in Table 2 shows that, for the given compression depth of 16~24 pixels, the reconstructed characterizing feature depth (step depth of ➀ and slot depths of ➁➂➃, by solving for the depth difference between the two blue + positions in Figure 7b) is 8.40~12.89 pixels, and the RMSE of the result by the proposed method within the contact area is in the range of 0.79~1.83 pixels, away from the result of the traditional benchmark. The results clearly indicate the capability of our method with a considerable reconstruction accuracy compared to the traditional look-up table method. However, the implementation process is much simpler, the proposed depth perceptual method is mathematically formulated with a robust depth perception precision, and it requires only one image for calibrating the sensor lighting condition, free of data-expensive calibration or learning procedures.

### 3.2. Pose Estimation for a Grasped Target

The test platform depicted in Figure 6 was utilized in a set of compression tests in which the poses of the targets were estimated. These tests aimed to mimic the application situation in which a VTS is attached to a robot arm’s fingertip in manipulating or grasping some objects, see Figure 1a. A capability of the VTS in estimating the pose of the object in contact is of a great desire by the manipulator for its decision and control in such a common operation. Several sets of contact targets were applied, in which each set consisted of two targets of the same geometry but different poses in a contact test. To test the pose estimation capability of the prototype with the perceptual scheme proposed, one target in a set was selected as of the reference pose. The relative pose of the other target to the reference was to be estimated.

In order to find a relative pose estimation, a point cloud registration technique is leveraged seeing the fact that a reconstructed depth map is essentially a point cloud. Once two point clouds characterizing the contact of the reference target and that of the to-be-estimated target are obtained individually from the reconstructed depth maps, the point cloud of the to-be-estimated target is registered against that of the reference one. A transformation connecting the reference with the to-be-estimated pose could be identified by minimizing the point cloud registration error. Both relative orientation and translation are contained in the transformation so that the actual relative pose of the to-be-estimated target to the reference could be fully determined.

The tests were conducted with the help of library *PCL 1.12.1* in Visual Studio 2019. The depth perception using the model-driven method proposed was employed. The points in the resulting depth map were first de-noised applying a prescribed threshold by which a tiny fluctuation in depth values was smoothed out. A contact area was identified in the de-noised depth map and all non-contact pixels were set to zero. The cleaned depth map was then down-sampled into a representative point cloud which is sparser than the depth map for the sake of processing speed. After representing the representative point cloud in the corresponding contact coordinate system O_c_X_c_Y_c_Z_c_, the point cloud was conversed into the *pcd* format. A point cloud registration was then conducted, in which Sample Consensus Initial Alignment (SAC-IA) [26] was employed as a coarse phase and Iterative Closest Point (ICP) [27] as a refined phase. The registration yields both the rotation matrix and the translation vector of the to-be-estimated point cloud in terms of the prescribed reference pose of the target.

Three sets of targets were 3D-printed then tested with the sensor prototype on the platform for the pose estimation. The targets, as depicted in Figure 6b, consist of I. a tool handle, II. a bottle cap, and III. a “HUST” stamp. There was a pin-hole on the assembling tail of each target for facilitating the pin attachment of the target to the presser rammer. The pin-holes vary in direction so that a unique contact pose is endowed to a target once it is attached to the flange. The axial direction of a pin-hole is indicated by a red arrow in Figure 6b. For each set of targets, the one on the left prescribes the reference pose, and the pose of the right one is to be estimated. The testing configuration in which a target was firmly assembled to the platform is shown in Figure 8a. For each target, the compressing tester was commanded to extend the rammer to a given displacement causing a contact between the target and the VTS. The displacement for contact is constant within each set of targets. The contact images obtained that way are presented in Figure 8b. The point cloud from the reconstructed depth map was then de-noised, contact-area-extracted and down-sampled, with the number of points reduced from around 5 × 10^5^ to around 10^3^. For the down-sampled representative point clouds, see Figure 8c. Figure 8d provide the final outcome of the point cloud registration, where red points denote the reference pose, and blue ones denote the registered pose of which the rotation and translation are both to be determined. Of the relative orientation angle and the translation achieved in the pose estimation, the truth, the estimated, the error, the result RMSE, and the time consumption of registration, are all tabulated in Table 3.

The following observations can be made from the experimental results in Figure 8 and Table 3.

In comparison with the known ground truth specified by the target design, the errors of the estimated overall relative orientation angles of cases I, II, and III, in a spatial view, were 0.6430°, 2.3461°, and 2.5936°, respectively. Additionally, the errors of translation displacement all fell within 1mm. The RMSEs of the registered point cloud against the reference one in all cases were quite restricted to a sub-millimeter level, indicating a perfect coincidence obtained by the registration process. We could tell from the errors in both orientation and translation together with the registration RMSE that the estimation of the pose of each target is fairly accurate.In each depth map, partial points near the edge of the contact area were removed in de-noise before building up the point cloud. That way, the number of almost-zero points was reduced in registration as far as possible for a reliability concern while trying to keep the shape reconstructed intact. For instance, the letters “HUST” in case III form a multi-connected domain in a mathematical sense. In handling this case, some points with an almost-zero depth value were treated as noise-contaminated and thus neglected in building up the point cloud, making the left portion of the letter “H” shortened a little bit.The time consumption for each registration round was about 6~8 s in our experiments, as the number of points in the cloud was relatively large at around 10^3^. Some advanced procedures such as discrete sampling could be expected in our future work in order to speed up the registration.

## 4. Conclusions

A novel model-driven tactile geometry perception methodology adapted for VTS is presented in this paper, for which an end-to-end 3D contact deformation depth perceptual scheme driven by a near-field photometric stereo model is realized. The perceptual scheme presented is essentially a combination of an offline illumination calibration with an online reconstruction. The identified illumination parameters are kept offline in the database for use in the online matrix calculation conducting the reconstruction. Compared to the existing data-driven method, this proposed scheme requires only one image, taken in a non-contact free situation, for calibrating the sensor lighting condition, free of data-expensive calibration or learning procedures. In addition, the non-uniformity of the illumination condition is embodied by the stereo model, resulting in a robust depth perception precision. Two sets of simulations were conducted by which the model-driven method presented was quantitatively validated.

Real contact tests were conducted on the designed VTS prototype. Taking a traditional look-up table method as the benchmark, the proposed model-driven depth perceptual scheme has shown its capability in figuring out a 3D contact shape of a compatible high quality. By contacting some target parts manually onto the sensor, both geometry shape and sub-millimeter texture detail has been covered by the proposed method with a good satisfaction. The overall result RMSE within the contact area was no greater than 100 μm, indicating a remarkable reconstruction precision. Additionally, a test platform was established for testing the target pose estimation capability of a VTS with the method proposed.

Several sets of contact targets were applied, in which each set consists of two targets of the same geometry but different poses in a contact test. The associated tests on three cases yielded pose estimation errors that were all less than 3° in terms of spatial orientation and all less than 1 mm in terms of translation. The accuracy of the pose estimation method proposed has thus been proved.

On the basis of this paper, the software and hardware integration technology of VTS can be further developed in the future. In the practical implementation, the sensor is installed on a robot arm’s fingertip to realize the contact perception of grasping and dexterous operation. Based on the rapidly reconstructed depth information, the robot can quickly obtain the geometric features and texture of the grasped object, identify defects such as scratches and cracks on the surface of rigid bodies, and even estimate the pose of the grasped instrument during dexterous operation. Furthermore, the incorporation of vision- and tactile-based sensing is urgently needed. The proposed method can be developed to assist vision-based depth perception for contact location and pose perception, especially in the occluded area, and has the potential to help robots better perform grasping or manipulation tasks.

## Figures and Tables

**Figure 1 sensors-22-06470-f001:**
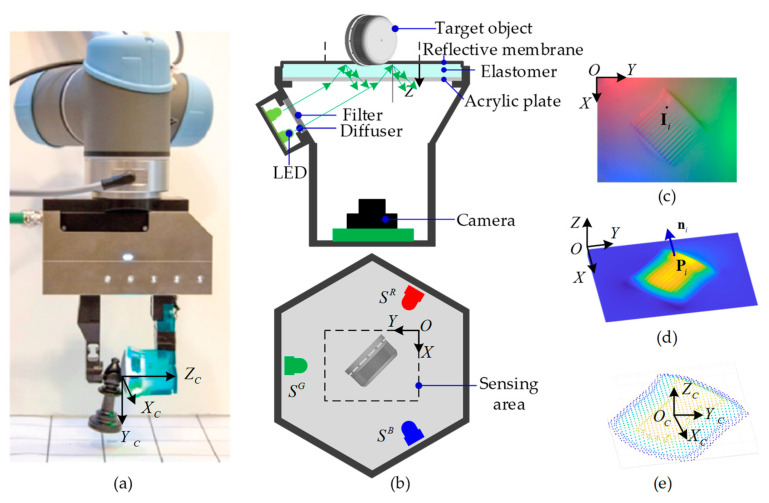
The VTS and its principle: (**a**) A manipulator operating with a VTS on its fingertip; (**b**) Configuration of a VTS (side view and top view); (**c**) Contact image captured; (**d**) Depth map reconstructed; (**e**) Point cloud characterizing the contact.

**Figure 2 sensors-22-06470-f002:**
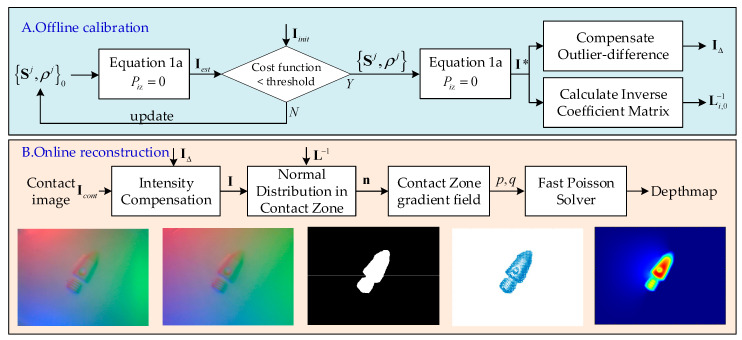
Flowchart of the model-driven depth perceptual scheme proposed for VTS.

**Figure 3 sensors-22-06470-f003:**
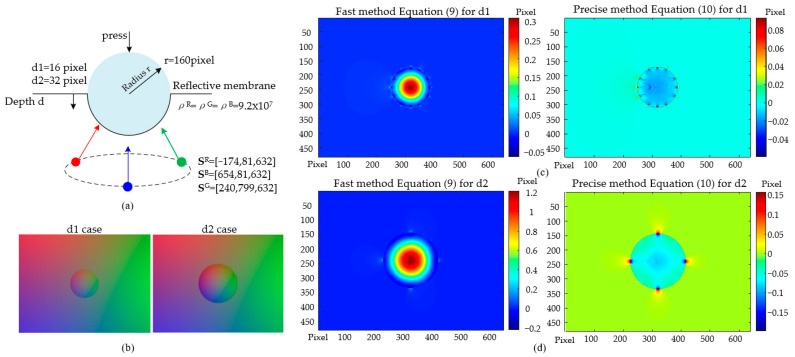
Reconstruction based on a simulation of the near-field photometric stereo model: (**a**) Compressing sphere; (**b**) Contact images of shallow-press (d1) and deep-press (d2); (**c**) Reconstruction error in the d1 case; (**d**) Reconstruction error in the d2 case.

**Figure 4 sensors-22-06470-f004:**
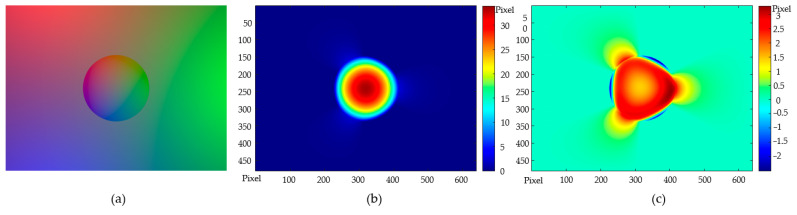
A sphere contact perception based on TACTO: (**a**) Contact image; (**b**) Reconstruction result utilizing the two-step method; (**c**) Reconstructed depth error.

**Figure 5 sensors-22-06470-f005:**
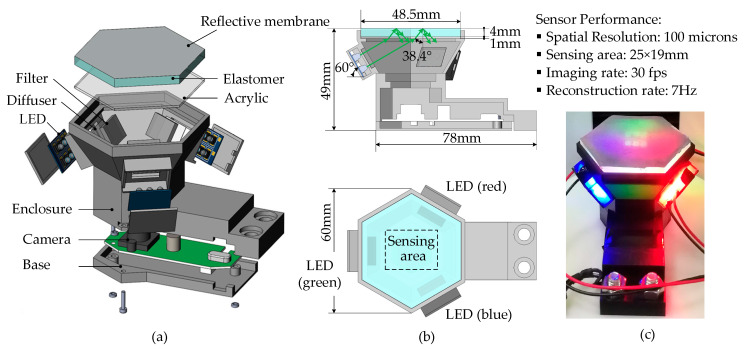
Design of the VTS: (**a**) Exploded views of the assembly; (**b**) Overall dimensions and the light path scheme; (**c**) Prototype with its performance briefed.

**Figure 6 sensors-22-06470-f006:**
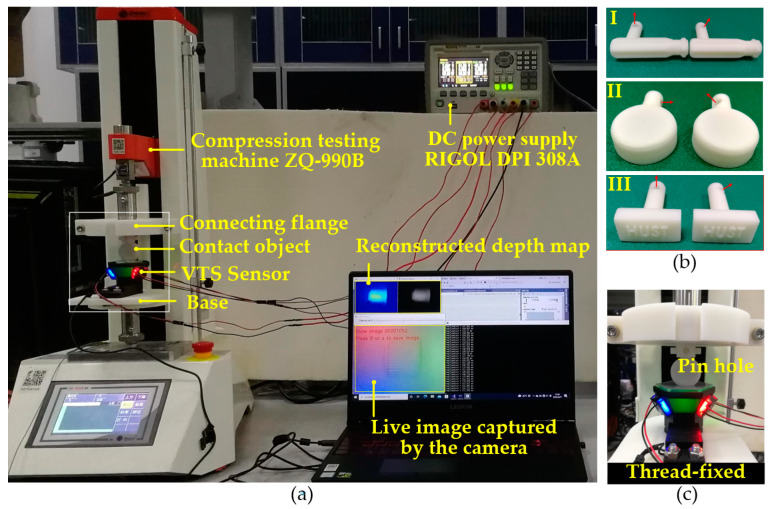
Test platform for VTS: (**a**) Overview; (**b**) Three sets of contact targets; (**c**) Detailed view of a contact.

**Figure 7 sensors-22-06470-f007:**
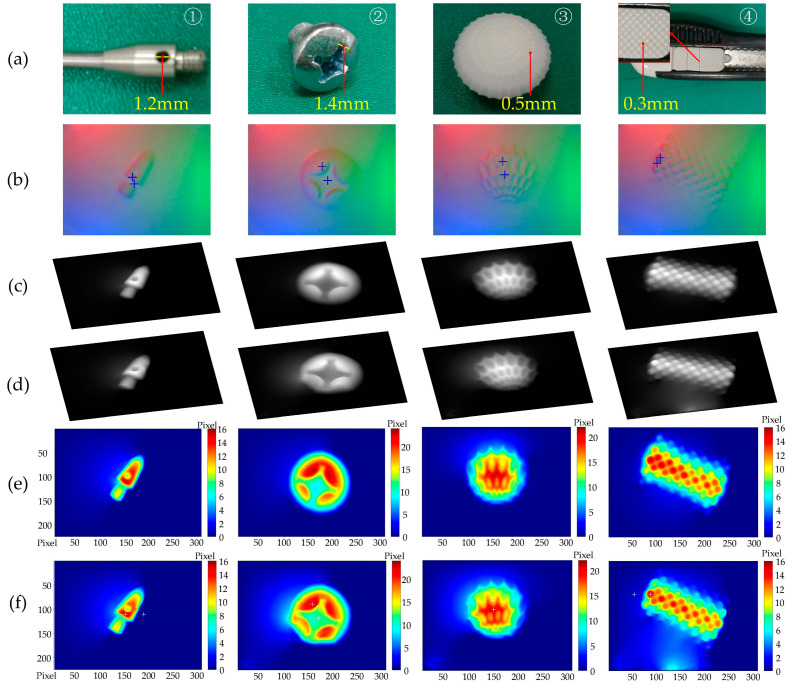
Results of depth perception by two methods using VTS: (**a**) Target; (**b**) Contact images where two blue + are labeled for characterizing feature depth calculation; (**c**) Result by the model-driven method proposed; (**d**) Result by the look-up table method; (**e**) Colorized result in (**c**); (**f**) Colorized result in (**d**).

**Figure 8 sensors-22-06470-f008:**
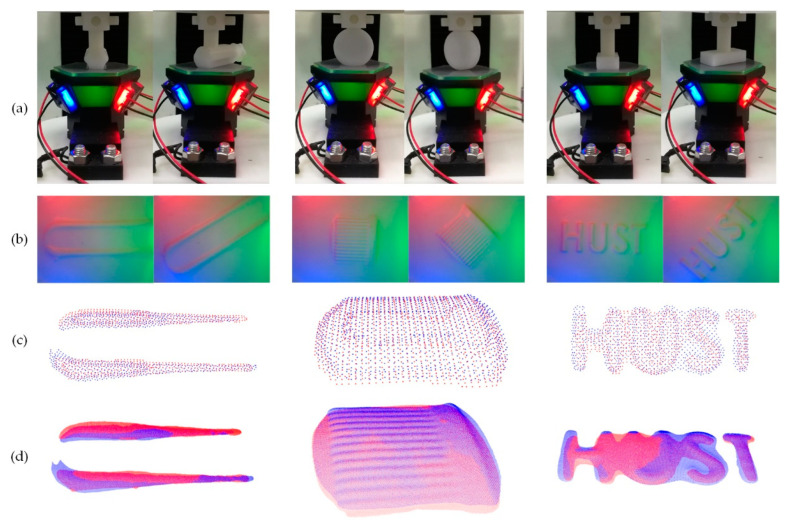
Point cloud registration over three cases for pose estimation: (**a**) The testing configuration; (**b**) The contact images; (**c**) The down-sampled representative point clouds; (**d**) The final outcome of the point cloud registration.

**Table 1 sensors-22-06470-t001:** Material and source of VTS Components.

Component	Material/Model/Property	Fabrication/Source
LED (red)	LR T64F-CBDB-1-1-20-R33-Z (wavelength = 625 nm)	ams-OSRAM AG
LED (green)	LT T64G-EAFB-29-N424-20-R33-Z (wavelength = 532 nm)	
LED (blue)	LB T64G-AACB-59-Z484-20-R33-Z (wavelength = 469 nm)	
Acrylic Plate	Polymethyl methacrylate	Laser cutting
Elastomer	Solaris A:Solaris B:Slacker = 1:1:1	Cold mold
Reflective membrane	Aluminum spherules and protective silicone rubber	Manual coating
Camera	C310 USB web-camera (resolution = 480 × 640)	Logitech^®^, Switzerland
Filter	Transmittance = 25%	PHTODE^®^, China
Diffuser	LGT125J, a PET film (Transmittance = 66%, Haze = 95%)	Commercially available
Enclosure	Black resin	3D-printed

**Table 2 sensors-22-06470-t002:** Difference between results of look-up table method and proposed model-driven method (in Pixel, 1 pixel = 39.5 μm).

Target	➀	➁	➂	➃
RMSE over the contact image	0.26	1.28	1.12	1.24
RMSE over the contact area	0.79	1.83	1.73	1.17
Maximum compression depth	16	24	22	16
Reconstructed characterizing feather depth	8.40	12.89	9.12	8.68

**Table 3 sensors-22-06470-t003:** Pose estimation errors in relative orientation and translation, registration RMSE, and time consumption for registration.

Cases	Item	Relative Orientation (°)	Translation (mm)	Registration RMSE (mm)	Time Consumption (s)
I. Tool Handle	Ground truth	(0, 0, −30)	(0, 0, 0)	0.1540	6.305
	Estimated	(0.2423, −0.0494, −29.3991)	(0.1303, 0.532, −0.0128)		
	Overall Error	0.6430°	(0.1303, 0.532, −0.0128)		
II. Bottle Cap	Ground truth	(0, 0, −45)	(0, 0, 0)	0.0619	8.206
	Estimated	(0.4354, −0.3108, −47.2867)	(−0.6321, 0.7604, 0.0158)		
	Overall Error	2.3461°	(−0.6321, 0.7604, 0.0158)		
III. “HUST” Stamp	Ground truth	(0, 0, −50)	(0, 0, 0)	0.1827	8.024
	Estimated	(0.3396, 0.5836, −52.5028)	(−0.1980, 0.2841, 0.0339)		
	Overall Error	2.5936°	(−0.1980, 0.2841, 0.0339)		

## Data Availability

Not applicable.

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
