# Peer review of "Model-Based 3D Contact Geometry Perception for Visual Tactile Sensor"

_sensors, 2022, doi:10.3390/s22176470_

Round 1
Reviewer 1 Report
The paper is interesting and well written. I do not see particular flaws in the methodology and in the results.
I suggest to be more effective in stating the main aims of the work.
Author Response
Dear reviewer:
We take this opportunity to thank you for taking the time to review our paper and raising compliments on our contributions. The detailed response please find in the attached PDF file.
Thanks again for your efficient handling.
Yours sincerely.
Jingjing Ji

Reviewer 2 Report
Thanks for the opportunity to review this study. I very much agree with the importance of this direction and its due contribution. However, I have the following considerations that I would like the authors to reconsider
(1) Overall, the novelty of this article is not known from the discussion gap between the conclusion and the existing literature, and it is suggested that the authors need to greatly strengthen this aspect.
(2) Experimental and practical implementations need to be able to be presented in the context of conclusions and future directions.
(3) The reason and process of "Three-Dimensional Geometry Perceptual Scheme" are not clear. It is a figure that appears suddenly. It is suggested that the authors need to strengthen the required statements.
Author Response
Dear reviewer:
We take this opportunity to thank you for taking the time to review our paper and raising your constructive suggestions. The detailed response please find in the attached PDF file.
Thanks again for your efficient handling.
Yours sincerely.
Jingjing Ji

Reviewer 3 Report
This is a well-written nice paper. Their new method, visual-tactile sensor (VTS), is well explained. I know virtually nothing about tactile sensor per se and I know nearly nothing about robotics. When the editor assigned this manuscript to me, I was hesitant but somehow I said yes. I regretted but even so, I found reading this paper was enjoyable. The paper is well-written and thorough. I didn’t find any strange expressions, logical gaps, or lack of explanation per se. I just have three concerns.
1) My first concern is about the verification experiments they conducted. The validity of their VTS method almost entirely rests on the comparison they made. In Experiment 1, the model performance was measured by RMSE where the ground truth was the estimations made by the look-up table method. Basically, what they showed (or verified) was how far away their model deviated from the traditional look-up table method. For me, this verification method is questionable. They explain why this comparison was made (Lines 301-308; 434-438. “the difficulty in getting the true value of depth in this manual test circumstances... “). But how do the authors know their VTS method is better than other traditional models? In Experiment 2 (3.2 Pose estimation for a Grasped Target), the authors introduce 3-D printed objects. Here, I assume the ground truth was known. But even here, no between-model comparisons were made. Did the VTS outperform other competing models (e.g., the traditional “look-up table” method)? If so, how did they know?
(2) It seems odd for me to call their method “tactile detector”. The simulation and other examples (Figure 7) appear barely related to “tactile” detection in the sense that these objects have clear “depth” information. These objects are also well suited for vision-based depth sensing (these objects reflect light readily). What if they choose more challenging objects (or surface), e.g., leather, wood, rocks, etc.? Does their method work well in more challenging contexts?
(3) What are the limitations of their method? What are pros and cons of their method? Does this method work well when vision-based depth perception is difficult. It is possible that an image can be degraded or occluded. How does the proposed model handle these conditions?
Author Response

(The authors gave the same response as above.)
